# Ecological Requirements for Abundance and Dispersion of Brazilian Yellow Fever Vectors in Tropical Areas

**DOI:** 10.3390/ijerph21050609

**Published:** 2024-05-10

**Authors:** Amanda Francisco Prado, Paula Ribeiro Prist, Luis Filipe Mucci, Patrícia Domingues de Freitas

**Affiliations:** 1Department of Genetics and Evolution, Center for Biological and Health Sciences, Federal University of São Carlos, Rodovia Washington Luis km 235, São Carlos 13565-905, SP, Brazil; patdf@ufscar.br; 2EcoHealth Alliance, 520 8th Avenue ste 1200, New York, NY 10018, USA; prist@ecohealthalliance.org; 3Taubaté Regional Lab., State Department of Health of São Paulo, Instituto Pasteur, Pça. Coronel Vitoriano, 23, Taubate 12020-020, SP, Brazil; lfmucci@gmail.com

**Keywords:** Atlantic Forest, disease ecology, human health, virus, land-use change, landscape ecology, abundance modeling, risk area, connectivity

## Abstract

In the Americas, wild yellow fever (WYF) is an infectious disease that is highly lethal for some non-human primate species and non-vaccinated people. Specifically, in the Brazilian Atlantic Forest, *Haemagogus leucocelaenus* and *Haemagogus janthinomys* mosquitoes act as the major vectors. Despite transmission risk being related to vector densities, little is known about how landscape structure affects vector abundance and movement. To fill these gaps, we used vector abundance data and a model-selection approach to assess how landscape structure affects vector abundance, aiming to identify connecting elements for virus dispersion in the state of São Paulo, Brazil. Our findings show that *Hg. leucocelaenus* and *Hg. janthinomys* abundances, in highly degraded and fragmented landscapes, are mainly affected by increases in forest cover at scales of 2.0 and 2.5 km, respectively. Fragmented landscapes provide ecological corridors for vector dispersion, which, along with high vector abundance, promotes the creation of risk areas for WYF virus spread, especially along the border with Minas Gerais state, the upper edges of the Serra do Mar, in the Serra da Cantareira, and in areas of the metropolitan regions of São Paulo and Campinas.

## 1. Introduction

Wild yellow fever (WYF) is an arbovirus disease caused by *flavivirus*, which has high lethality rates in Brazil [1,2,3,4]. *Haemagogus* spp. and *Sabethes* spp. mosquitoes are its main vectors, whereas non-human primates (NHP) act as amplifier hosts [5,6,7]. WYF was considered endemic to the Amazon region; however, in recent decades, the virus expanded to other regions of Brazil, especially to the southeast and south regions [1,8,9], which include areas of the Atlantic Forest and the Cerrado, biomes that have been highly altered due to habitat loss and fragmentation caused mainly by anthropogenic activities [10,11,12,13].

The factors related to the re-emergence and spread of the virus to areas that had for a long time been considered unaffected are still poorly understood, but it is hypothesized that deforestation, habitat fragmentation [14,15,16,17,18] and climate change [15,19,20,21] may affect the extrinsic incubation period, abundance and vector spatial distribution patterns [15,22,23,24]. Specifically, infected mosquitoes could be using stepping stones, forest corridors, and/or forest roads to disperse to new areas [17,25,26], where they could find susceptible NHP populations, leading to new WYF outbreaks [6,21,23].

The largest and most recent outbreak of WYF recorded in Brazil occurred between 2016 and 2021 [9,15,27] in the southeast and south regions of the country [4,27,28,29]. In these regions, *Hg. leucocelaenus* and *Hg. janthinomys* are the main WYF vectors [30]. Both species are diurnal, common, and dependent on forest habitats, particularly for reproduction, as they lay eggs in tree and bamboo holes [31,32]. However, they can differ in their tolerance to anthropogenic habitat modifications, with *Hg. leucocelaenus* showing greater adaptability to disturbed and degraded environments [32,33,34], whereas *Hg. janthinomys* tends to establish itself in more humid and preserved forests [35]. *Hg. leucocelaenus* and *Hg. janthinomys* have extensive flight capabilities, being able to travel, respectively, up to about 6 and 11 km over their lifetimes, even in deforested areas [5,31,36].

As the virus spread is more likely to occur through the dispersal of infected vectors [15,17,36], it is essential to understand how changes in the landscape structure can affect their abundances and movement behaviors, modulating disease risk. By taking this approach, it would be possible to predict when land-use changes would lead to increases in vector abundance and facilitate vector movement to other regions, and to establish preventive measures before the onset of epizootic events. Because WYF is not treatable, the main prophylactic measure is the vaccination of people heading to risk areas, in addition to individual protection to avoid contact with mosquitoes and population control of WYF vectors [1,37].

The objectives of this study were to analyze, through a multiscale perspective, how landscape structure affects the abundance of *Hg. leucocelaenus* and *Hg. janthinomys* in the Atlantic Forest of São Paulo state, and to identify the landscape features that allowed vector dispersion throughout the state during the 2016–2021 outbreak. Our hypotheses are as follows: (i) *Hg. leucocelaenus* and *Hg. janthinomys* abundance are positively affected by forest cover; (ii) the scale of effect for *Hg. janthinomys* is bigger than for *Hg. leucocelaenus*, as it is related to species dispersion capabilities [38]; and (iii) more conserved and more degraded landscapes can limit virus spread, either because the distance between these fragments is too great to allow dispersion, or because the landscape is intact and very diverse in terms of shelter and food, diluting the effect of the virus on populations.

## 2. Materials and Methods

### 2.1. Study Area and Data Collection

Abundance mosquito data and geographical coordinates of sample points were obtained in Mucci et al. [39,40], where *Hg. leucocelaenus* and *Hg. janthinomys* abundances were collected every three months, between October 2006 and July 2008, at 24 points located in 22 municipalities of the administrative regions of Araçatuba and São José do Rio Preto, in the northwest region of the state of São Paulo (Figure 1). The predominant climate of this region is Cwa, according to the Köppen classification, with a rainy summer and a dry winter. Due to the impossibility of morphologically distinguishing females of *Hg. janthinomys* and *Hg. capricornii* [41], they were treated together as *Hg. janthinomys*/*capricornii*.

Mosquitoes were collected using hand-held nets and mouth aspirators (IEC capturers) at ground level along 1.5 km long trails (from the edge to the center of forest fragments). At each point, mosquitoes were captured over two consecutive days, between 9:00 a.m. and 3:00 p.m., with walking segments inside the forests, including multiple stops to replicate the intrusion effect. Further details regarding the study area and data collection procedures are described in Mucci et al. [39,40].

### 2.2. Landscape Metrics and Predictor Variables

To determine the scale of effect [38] for *Hg. leucocelaenus* and *Hg. janthinomys*/*capricornii* abundance in the Atlantic Forest of São Paulo, the landscape structure was analyzed using a multi-scalar approach [42]. Scales were defined as concentric circles around each sampling point, with radii of 500 m, 1 km, 1.5 km, 2 km, and 2.5 km. Those distances were based on the dispersal distance observed for vectors during summer and winter (200 m to 1.2 km/day, respectively [17]) and on the assumption that the scale of effect ranges between four to nine times the average dispersal distance for each species [38]. The maximum distance was also established to avoid overlap among the different scales (namely landscapes), as the minimum distance between two collection points was approximately 2.35 km.

Landscape metrics at each scale were calculated based on the “Projeto Mapbiomas” collection 6 of the Annual Series of Land Cover and Use Maps of Brazil, which utilizes Landsat image mosaics with a 30 m resolution [12]. The mapping used was from the year 2007 to coincide with land use and land cover during the collection period (2006 to 2008). Selected metrics for each landscape were (i) the percentage of forest cover; (ii) the percentage of total native vegetation (the sum of both forest and savanna-formation percentages); (iii) the percentage of agricultural land; (iv) the number of forest fragments divided by the landscape area (km^2^); (v) the number of native-vegetation fragments divided by the landscape area (km^2^); (vi) forest edge density; (vii) native-vegetation edge density; (viii) cohesion index among forest fragments; and (ix) cohesion index among native-vegetation fragments. All analyses were performed in ArcGIS 10.8 and Fragstats 4.2.1.

### 2.3. Model Selection for Landscape Structure and Scale of Effect

To analyze the influences of landscape variables and the scale of effect on the abundance of *Hg. leucocelaenus* and *Hg. janthinomys*/*capricornii*, a model-selection approach [43] was performed for each species separately. Abundance data for each species were normalized with log + 1, and generalized linear models (glm function with normal distribution, stats package, RStudio software [44], version 2022.01.1) were fitted and compared.

For each species, four null models were developed; these included a model with abundance varying randomly not being affected by our predictor variables. The other three geographical null models considered the spatial location as more important than the predictor variables, with abundance data being affected (i) only by latitude; (ii) only by longitude, or (iii) by both latitude and longitude. In addition to these models, for each scale, simple models were developed with each predictor variable, except for the metrics related to cohesion index, which do not make sense alone in the model; and additive models were also developed with 2 or 3 variables. Every model had a hypothesis of its effect on the species’ abundance. We did not use the variables of percentage of forest cover and percentage of agricultural lands in the same model, as the sum of these percentages at some landscapes was close to 100%; we also did not include correlated variables with a correlation score above 0.7. In addition, as forest and native-vegetation metrics were related to different biomes and native-vegetation formations, they were never included together in the same model. In that way, we could also test which formation type (Atlantic Forest or Cerrado formations) had a greater effect on vector abundance. Considering all scales, a total of 109 competing models were developed for each species (Appendix A).

Model comparison was performed using the MuMIn package in the R software [45] and the Akaike Information Criterion corrected for small samples (AICc [43]), considering models with ΔAICc ≤ 2 equally plausible [43]. The relative importance of each scale was calculated from the sum of the weights of evidence of all models that shared that scale [46]. The same process was performed to calculate the relative importance of each variable present in the selected models, but the sum was weighted by the total number of models in which that variable was present.

We also calculated the evidence ratio of each model, scale, and variable, which refers to the ratio between the weight of evidence of the best model, scale, or variable compared to those of the others. This indicates how many times better the first factor is at explaining a certain pattern than are the other factors [46]. The slope and the standard error for each variable were calculated from its univariate model. To detect potential spatial autocorrelation, we computed a Mantel test based on 5000 permutations using the straight-line distances among the sampled landscapes (measured from the central point of the landscape unit) and the residuals of the best models, confirming that there was no significant spatial autocorrelation (Mantel test, r = −0.09, *p* = 0.81 for *Hg. leucocelaenus* and r = −0.04; *p* = 0.62 for *Hg. janthinomys*/*capricornii*).

### 2.4. Extrapolation of Vector Mosquito Abundance

Based on the results of the model-selection procedure, and on the assumption that vector species are associated with natural vegetation environments and are not commonly found in other types of environments such as agricultural areas and urban areas [47,48,49,50], *Hg. leucocelaenus* abundance was extrapolated to forest fragments of São Paulo state, whereas *Hg. janthinomys*/*capricornii* abundance was extrapolated to native-vegetation fragments (forest along with savannah formation).

For the vector abundance extrapolation, the parameters of the best selected model for each species were calculated and used. As the concurrent models were normalized through the *log* + 1 of abundance, the inverse function of the best model equation was used for the calculation of extrapolated abundance. In other words, we used the equation *y = exp* (*a* + *bx*) − 1, where *y* is the extrapolated abundance for the species (per pixel), *a* is the intercept, *b* is the slope, and *x* is the per-pixel value of the predictor variable.

For both species, extrapolations followed the range found for all the predictor variables. In that way, if the minimum and maximum values of forest cover present in the sampled landscapes were, respectively, 0% and 50%, and this variable was selected in the best model, then extrapolations were made only to areas within this range. The extrapolation of abundances was performed in ArcGIS 10.8 software, using the land-cover and land-use map from the “Projeto Mapbiomas” (collection 6) for the year 2016, to coincide with the functional connectivity analysis and corridors for the vectors described below.

### 2.5. Ecological Corridors and Risk Areas for Yellow Fever Spread

To identify potential forest corridors and stepping stones that may have allowed dispersion of the WYF vectors, we performed a graph analysis based on vector ecological requirements (Conefor Inputs, 1.0.218). Only the type of habitat present in the selected models for each species was included—forest fragments for *Hg. leucocelaenus* and native vegetation (forest along with savannah fragments) for *Hg. janthinomys/capricornii*. Then, a functional connectivity map for each vector was generatedconsidering, for both species, that only fragments located within 1 km of each other would be functionally connected. This distance was chosen because it is the average daily virus propagation distance by vectors in São Paulo state, as evidenced by recent studies [17,20,24,28,51].

Additionally, as *Hg. janthinomys*/*capricornii* showed a stronger association with interior areas of native-vegetation fragments, the calculation of the aforementioned distance (1 km) was performed from the centroid of the habitat fragment, whereas for *Hg. leucocelaenus*, the calculation was performed from the edge of the fragment. For this analysis we used “Projeto Mapbiomas” (collection 6) for the year 2016, which is when the virus managed to spread throughout the state.

From the species-functional-connectivity map, we calculated the density of connections within a 1 km radius, i.e., the number of connections/km^2^, resulting in a statewide map of potential functional connectivity for each species. Each map of potential functional connectivity was overlaid with the extrapolated abundance map of the corresponding species. By combining these two parameters, it was possible to determine which of the areas of São Paulo state are capable of harboring higher vector abundances and providing high functional connectivity for their dispersal, i.e., areas of high risk for vector dispersion and, consequently, virus and disease spread.

## 3. Results

### 3.1. Relationship between Species Abundance, Landscape Structure and Scale of Effect

During the study period, 345 individuals of *Hg. leucocelaenus* were captured (mean = 14.37 ± 27.56) in 66% of the total sampling points. *Hg. janthinomys*/*capricornii* was collected in 33% of these points, totaling 147 individuals (mean = 6.12 ± 22.40), with approximately 75% of these having been collected at a single location (Figure 1).

Our findings pointed out that six equally plausible models explained the abundance of *Hg. leucocelaenus* (Table 1), with the amount of forest cover (in different spatial scales) being the most important variable (Table 2) and appearing in five of the models (Table 1), consistently having a positive effect on mosquito abundance (Table 2). The fifth model also included the number of forest fragments per km^2^ at the 1 km scale (Table 1), which showed a non-significant effect (Table 2); and the sixth model included the sum of native-vegetation percentage at 1 km scale, which positively influenced the species abundance (Table 2), and the number of native-vegetation fragments per km^2^ (Table 1), which showed a non-significant effect (Table 2). The strongest scale of effect for this vector was 1.0 km, albeit with a low evidence ratio, indicating that the other scales could also explain species abundance, with the exception of the 500 m scale for which the cumulative-evidence weight was less than 0.001 (Table 3).

For *Hg. Janthinomys/capricornii*, four models were selected as plausible to explain the species abundance, all of which contained at least one landscape-composition variable (percentage of forest cover, agricultural land or native vegetation), and two also contained edge density for forest or native-vegetation fragments (Table 1).

The variable with the strongest effect to explain *Hg. janthinomys*/*capricornii* abundance was the percentage of forest cover at the 2.5 km scale, followed by the forest edge density and the percentage of agricultural land, which were both also within a 2.5 km radius (Table 2). Regardless of the scale, the percentage of forest cover and of native vegetation positively influenced species abundance, unlike the percentage of agricultural land and the edge density, which showed negative effects (Table 2). However, it is worth mentioning that the edge density did not show a significant result (Table 2). These results indicated that increases in native-vegetation cover and reduction in agricultural land in the landscape could positively influence *Hg. janthinomys*/*capricornii* abundance.

The species abundances responded to landscape structure mainly at larger scales, with the 2.5 km scale being two times better at explaining species abundance than the 2.0 km scale, which was the second best scale (Table 3). Overall, the sum of evidence weight for each scale decreased as the scale decreased (Table 3).

### 3.2. Vector Abundance Extrapolation, Corridors, and Risk Areas for Yellow Fever

The extrapolated abundance of *Hg. leucocelaenus* in the forest fragments of São Paulo state ranged from 0 to 70.44 individuals per pixel (Figure 2), with an average of 1.28 (±6.31) individuals per pixel. For *Hg. janthinomys*/*capricornii*, the extrapolated abundance for native-vegetation fragments in the state ranged from 0 to 12.84 individuals per pixel, with an average of 5.25 (±3.82) individuals per pixel (Figure 3).

For both species, the highest abundances were estimated in the upper and lower edges of the Serra do Mar that comprises much of the Vale do Paraíba, the metropolitan region of São Paulo, and the Vale do Ribeira, as well as in Serra da Cantareira, the southeast of metropolitan regions of Campinas and Ribeirão Preto, and the central region of the state, with the latter mainly for *Hg. janthinimys*/*capricornii* (Figure 2 and Figure 3). An interesting difference between the species is that low abundance for *Hg. janthinomys*/*capricornii* was estimated in the central area of the metropolitan region of São Paulo (Figure 3), which includes the city of São Paulo, whereas medium to high abundance was estimated for *Hg. leucocelaenus* in the same area (Figure 2); an opposite pattern was observed for the periphery of this region (Figure 2 and Figure 3).

The density of connections between habitat fragments varied from 0 to 737.81 connections/km^2^ for *Hg. leucocelaenus* (Figure 2), with an average of 7.26 (±20.94), and from 0 to 904.32 connections/km^2^ for *Hg. janthinomys*/*capricornii* (Figure 3), with an average of 7.87 (±25.34). The potential functional connectivity was quite similar for both species (Figure 2 and Figure 3). Overall, areas of higher connectivity were observed in the northeast, mainly in the border with the state of Minas Gerais, and in the central region of the state, where the potential functional connectivity for *Hg. janthinomys*/*capricornii* was slightly higher than that for *Hg. leucocelaenus*. High and medium connectivity were also observed in areas from the southwest to the east of São Paulo state, on the upper edges of the Serra do Mar, in areas of the Vale do Paraíba, and in peripheral areas of the metropolitan regions of São Paulo and of Campinas, including their surroundings such as the region of Serra da Cantareira. In general, low connectivity was observed in the western and northwestern regions of the state and in areas with a large amount of continuous forest, such as the core region of the Serra do Mar.

By overlaying the maps of extrapolated abundance and potential functional connectivity for each species, it was observed for *Hg. leucocelaenus* (Figure 2) that the areas at higher risk for yellow fever dissemination, which was due to them containing higher estimated abundance values with medium or high potential functional connectivity for the species, were concentrated mainly in the central and northeast regions of the state, especially on the border with Minas Gerais state, and from the southwest to the southeast of the state, on the upper edges of the Serra do Mar, in the Vale do Paraíba, in the region of Serra da Cantareira and its surroundings, and in the southeast of the metropolitan regions of Campinas and of São Paulo. The region with the lowest risk of dispersion for this vector was the western part of the state, which was due to low potential connectivity and/or low estimated abundance for the species for most of this region. For *Hg. janthinomys*/*capricornii* (Figure 3), a similar pattern was observed, but with higher risk in the central and northern regions of the state.

## 4. Discussion

### 4.1. Relationships between Species Abundance and Landscape Structure

We provide novel insights about how forest loss and the remaining landscape structure could be affecting the abundance and dispersion of the two main WYF vectors in São Paulo state: *Haemagogus leucocelaenus* and *Haemagogus janthinomys*. Our results showed that, for both species, increases in native-vegetation cover in highly degraded and fragmented landscapes could positively affect mosquito abundance and consequently influence WYF risk. At the same time, this degraded and fragmented pattern of the landscapes promoted ecological corridors for WYF virus spread.

For *Hg. leucocelaenus*, the percentages of forest and native-vegetation cover were the most important variables in explaining species abundance, consistently demonstrating positive effects. It is known that *Hg. leucocelaenus* has sylvatic habits, being dependent on forest environments to fulfill its ecological requirements such as feeding and oviposition [32]. Thus, it was expected that forested landscapes would harbor high abundances of this species, at least under certain conditions.

However, it is worth noting that this species exhibits ecological plasticity, being able to adapt well to areas that have a certain degree of degradation and anthropization [18,33,52,53,54]. As our study area is in a highly degraded region of Brazil, showing average forest-cover percentages between 15.6% (at a radius of 2.5 km) and 31.9% (at a radius of 500 m), and with a maximum of 63% of forest cover considering all landscapes, this result may depend on the low forest-coverage percentages found in the landscapes. In other words, it may be that there are non-linear responses, and that after a certain threshold of forest cover, the abundance of this species may have different responses. As an example, large areas of preserved and continuous forest, as found in the Serra do Mar, apparently does not harbor high abundance of this species [55,56,57,58,59,60,61,62,63,64].

There is still no knowledge about the existence of thresholds of forest cover percentage at different landscape scales beyond which there is an increase or decrease in the abundance of *Hg. leucocelaenus*. However, Ilacqua et al. [16] suggest that landscapes with intermediate percentages (from 30% to 70%) of forest cover are those with the highest risks of yellow fever re-emergence and occurrence, which may be related to vector abundance. Therefore, a plausible hypothesis is that most of our landscapes fell within the forest-cover range suggested by Ilacqua et al. [16], and so we only found a positive effect of forest cover on vector abundance. Thus, we suggest the development of studies that encompass landscapes at larger spatial scales (above 2.5 km) and also with forest-cover percentages above 70%, so that the hypotheses raised here can be tested.

Regarding *Hg. janthinomys*/*capricornii*, the species also showed a positive relationship with forest and native-vegetation environments, as expected due to its sylvatic habit [32]. Three out of the four selected models for the species included these variables, with the percentage of forest cover at 2.5 km radius being the most important variable, positively influencing abundance, followed by forest edge density, percentage of agricultural land, and native-vegetation edge density, which negatively influenced abundance. These results corroborated evidences that *Hg. Janthinomys* (i) is less abundant in more degraded and fragmented landscapes, which have less native vegetation and more edge areas [65]; and (ii) is more related to the interior areas of forest fragments or native-vegetation, establishing itself better in wetter [33,66,67] and more preserved areas [35,39]. These results also support evidences that this species has higher environmental requirements compared to *Hg. leucocelaenus* [32].

Despite *Hg. janthinomys* appearing to have secondary vectorial importance than *Hg. leucocelaenus* in the Atlantic Forest due to its seemingly lower abundance and distribution [2,30,33,34,35,47,52,53,54,59,68,69,70], it is known that the density, abundance, and distribution of *Hg. Janthinomys* can increase considerably during WYF-outbreak periods, as occurred in the southeastern region of Brazil during the 2016–2021 outbreak [30,50]. Additionally, due to their high susceptibility to the virus [1], these mosquitoes can reach high infection rates [30], especially in areas with less dense vegetation and lower mosquito diversity [30,50], which, combined with their primatophilic habit [39,71], makes them one of the most important yellow fever vectors in the biome, especially when they are in high abundance [30]. In fact, there was an apparent positive association between the abundance of the vectors studied here, particularly *Hg. janthinomys*/*capricornii*, with the epicenter of epizootics in the year 2008 [40,72].

Our results also showed a negative relationship between the abundance of *Hg. janthinomys*/*capricornii* and the percentage of agricultural land in the landscape, which was probably due to the lack of suitable conditions and necessary resources for the development and reproduction of these mosquitoes in these environments, including the absence of NHP and tree hollows for oviposition. Although it is known that the presence of livestock alters the structure of Culicidae communities [73], and also that *Hg. janthinomys* feeds on cattle blood [39,71], recent studies have shown that these areas also provide significant resistance to virus spread, probably due to the vector ecology [17,49], indicating that these mosquitoes do not use agricultural areas. However, few studies have sought to elucidate the effects of land-use changes, particularly the conversion of forested areas into agricultural areas, on the abundance of WYF vectors [74] and their consequent risks in the emergence, re-emergence, and incidence of arboviruses in human and animal populations.

Forest-edge areas interfacing with agricultural areas, however, facilitate virus spread by mosquitoes [17,49]. In recent decades, the main alterations in the Atlantic Forest biome have been related to the conversion of native vegetation into agricultural areas [12,13], which increases fragmentation, and consequently the amount of forest-edge areas in the landscape, while reducing the amount of core-native-vegetation areas [65]. Our results indicated that *Hg. janthinomys*/*capricornii* is more related to the interior of forest fragments. Nevertheless, it is known that such landscape alterations can impose adaptive pressures, including those based on host availability [75,76], which can lead to changes in aspects related to the niche of the mosquito vectors [39,77], thus forcing *Hg. janthinomys* to seek food at ground level and in more open areas, for example [32,39,71,78], where there are greater chances of contact with humans.

### 4.2. Scale of Effect

The strongest scale of effect to explain the abundance of *Hg. leucocelaenus* was 1.0 km, followed by 2.0 km, but there was a low evidence ratio for the former, indicating that the scale of effect should be between these values. Conversely, for *Hg. janthinomys*/*capricornii*, all selected models were at the scale of 2.0 or 2.5 km, with the 2.5 km scale being twice as good at explaining the species abundance as the second most important scale (2.0 km).

Jackson and Fahrig [38] suggest that the scale of effect for a species is 0.3 to 0.5 times its maximum dispersal distance. This distance is about 5.7 km for *Hg. leucocelaenus* [31,36] and 11 km for *Hg. janthinomys* [5,36]. Thus, the theoretical scale of effect for vectors would range between 1.7 and 2.8 km for *Hg. leucocelaenus* and 3.3 to 5.5 km for *Hg. janthinomys/capricornii*. However, because in this study it was not possible to test scales larger than 2.5 km due to the high overlap between landscapes, this may be the reason this scale was selected. Other studies should include larger scales and confirm the scale of effect for this species.

A study conducted in wetlands in Sweden found that Culicidae populations were influenced by forest cover within a radius of 3 km, which was the largest scale used in the study [79], and this is consistent with our results. On the other hand, another study conducted in the Brazilian Cerrado [22] found that the *Haemagogus* and *Sabethes* communities possibly have a relationship with forest cover in the immediate surroundings (100 m radius, the smallest used in the study). This may indicate one or more of the following: (i) the scale of effect for Culicidae populations varies according to the biome, as suggested by Alencar et al. [22]; (ii) the scale of effect proposed by Alencar et al. [22] was not accurately defined, being smaller than the range suggested by Jackson and Fahrig [38]; and/or (iii) environmental factors at both local scales [22,48,80,81] and broader scales [79] may affect the structure of Culicidae communities [76], modulating the relative abundances of different species.

### 4.3. Vector Abundance Extrapolation, Corridors and Risk Areas for Yellow Fever

As evidenced by our results, a large portion of São Paulo state provides potential functional connectivity for WYF vectors, forming several possible ecological corridors and stepping stones for mosquito dispersal and, consequently, virus spread. Associating this with the estimated abundance of both vectors, the areas identified as being at high risk of virus spread were, mainly, in the following areas: (i) the central and northeastern regions of São Paulo state, especially along the borders with Minas Gerais state; (ii) the southwest to east ranges, along the upper edges of the Serra do Mar and Vale do Paraíba; (iii) the Serra da Cantareira and its surroundings; and (iv) the southwest portions of the metropolitan regions of Campinas and of São Paulo.

Other studies that also mapped areas of risk or vulnerability for WYF indicated the southern to eastern strip of São Paulo state as areas with high [74,82] to moderate risk [82], including the region of the Serra do Mar. In the present study, the edges of the Serra do Mar, particularly the upper ones, showed a high risk for vector dispersal, corroborating studies indicating that edge areas are conducive for virus spread [17,18] and are associated with a higher risk of disease occurrence and re-emergence [16,18]. The core area of the Serra do Mar, however, was classified as low risk for WYF spread, due to low potential functional connectivity, based on vector ecology, that was observed for the region. It is known that core areas of forested regions can impose limitations on virus spread by mosquitoes [17,49]. However, one of our study limitations relies on the fact that the sampled landscapes had a low amount of forest cover. Consequently, we were unable to capture the effects of landscapes with higher cover on vector abundances and, as a consequence, extrapolation to high-cover areas would not be reliable.

The Serra da Cantareira and its surroundings also presented a high risk of WYF dissemination. The Serra da Cantareira, located in the north of the metropolitan region of São Paulo, harbors remnants of the Atlantic Forest in various stages of regeneration, which are considered the most important rainforest fragments for the city of São Paulo [83]. It was considered one of the main routes of virus entry into the city of São Paulo during the 2016–2021 outbreak [25,51,84]. In its surroundings are most of the municipalities with the highest number of epizootics and human cases that affected the state during this outbreak, such as Mairiporã, Atibaia, Bragança Paulista, Jundiaí, and Campinas [24].

One factor that draws attention is the high risk of WYF virus spreading near large urban centers, such as in some municipalities of the southeast of the metropolitan regions of São Paulo and of Campinas, which are the main population centers in São Paulo state. This poses a high risk of disease urbanization, as various other potential vectors of yellow fever, such as *Aedes scapularis* [47], *Ae. albopictus* [85], *Ae. aegypti* [86], and *Psorophora ferox* [47], as well as several species of NHP susceptible to the virus, are found in the forest remnants or in the urban green areas (e.g., parks) of these regions.

Areas that presented low risks of yellow fever dissemination were mainly the western and northwestern regions of the state and the central areas of the metropolitan regions of Campinas and of São Paulo, which is mainly due to the low potential functional connectivity found in these regions. In the western and northwestern regions of the state there is a small amount of native vegetation [10], as these areas have been mostly converted to agricultural areas over the years [13], and the remaining fragments have acquired a sparser configuration. The low connectivity observed in the central areas of the metropolitan region of Campinas and predominantly in the metropolitan region of São Paulo was mainly due to the high degree of urbanization of the area, which consequently left only a few, mostly isolated, forest fragments.

Recent studies have shown the importance of natural barriers, such as preserved native vegetation [17,24], mountain ranges like the Serra do Mar [29], and highly anthropized and modified areas such as agricultural areas and large urban centers [17,49], to limit virus propagation by restricting the movement of vector mosquitoes. The results obtained in this study corroborate these findings.

## 5. Conclusions

The Atlantic Forest remained free from yellow fever for at least six decades, experiencing cyclic reintroduction events for the virus from the 2000s. These events had significant impacts on human populations inhabiting the biome, particularly due to the absence of vaccine recommendations or low vaccine coverage, and they were devastating for various populations of NHP, some of which belonged to species threatened with extinction.

The re-emergence of yellow fever in the Atlantic Forest may have been caused by the changes that the biome has undergone in recent decades, in addition to large-scale climate changes. However, the studies regarding the impacts of Atlantic Forest land-use changes on the Culicidae mosquito community, especially on the populations of the main WYF vectors, are still in early stages.

The findings of this study shed some light on these issues. Overall, they indicate that increases in forest cover in highly degraded and fragmented landscapes may lead to an increased abundance of WYF vectors. Furthermore, fragmented landscapes are enabling vector dispersion and consequently leading to virus dissemination, thus boosting disease re-emergence and occurrence.

To prevent WYF spread, it is essential to include landscape analysis in the routine of epidemiological surveillance and animal health services to more accurately estimate potential virus-dissemination routes, considering a multiscale approach, i.e., at local, sub-regional, and regional levels, as each pattern of fragmented-landscape arrangement implies different possibilities, intensities, and speeds of virus spread in the extra-Amazon region, which is still considered non-endemic for WYF. The maps yielded in this study may guide the limited funds and efforts for these routines, for example, by directing researchers to important hotspots for mosquito collection and epidemiological analysis, which will help in establishing preventive measures before the onset of epizootic events.

## Figures and Tables

**Figure 1 ijerph-21-00609-f001:**
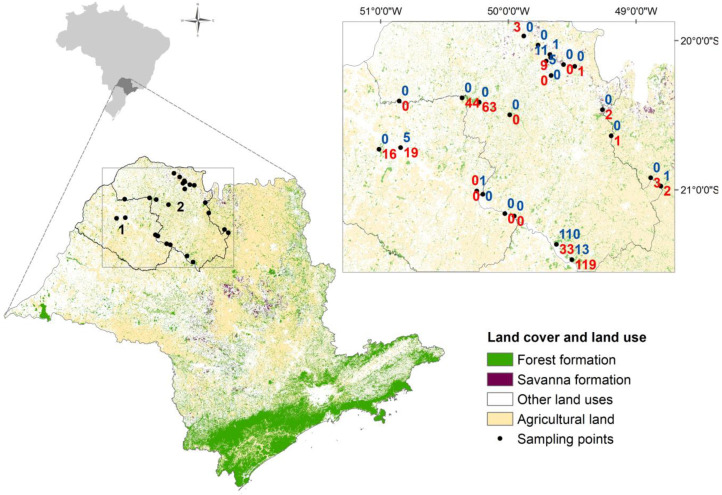
Mosquito-sampling points in the northwest region of São Paulo state. The rectangle on the right is a magnification of the region where mosquitoes were collected; numbers in red represent *Haemagogus leucocelaenus* abundances and numbers in blue represent *Haemagogus janthinomys*/*capricornii* abundances. Numbered regions correspond to the administrative regions of (1) Araçatuba and (2) São José do Rio Preto.

**Figure 2 ijerph-21-00609-f002:**
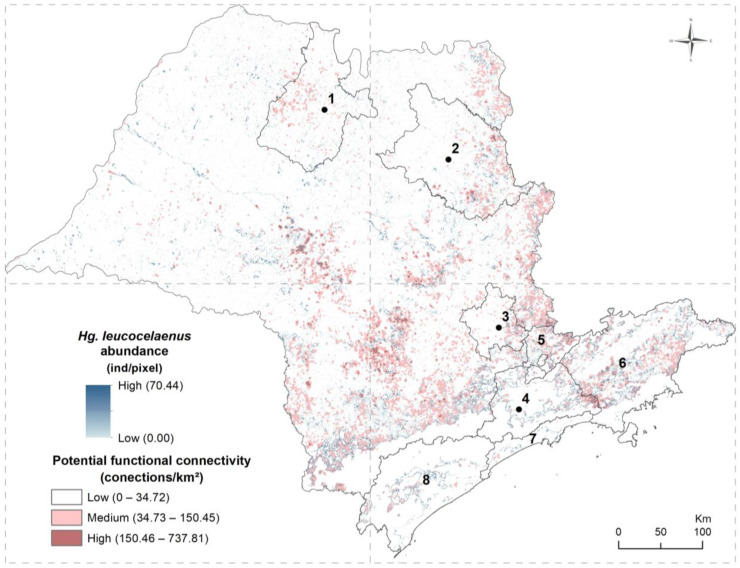
Estimated abundance and potential functional connectivity for *Haemagogus leucocelaenus* in the state of São Paulo, indicating risk areas for mosquito dispersion. Numbered regions (1–4) correspond to the metropolitan regions of (1) São José do Rio Preto, (2) Ribeirão Preto, (3) Campinas, and (4) São Paulo, with the point indicating the centroid of the city that gives its name to the region; (5) corresponds to Serra da Cantareira; and (6–8) correspond to administrative regions: (6) Vale do Paraíba and Litoral Norte, (7) Baixada Santista, and (8) Vale do Ribeira.

**Figure 3 ijerph-21-00609-f003:**
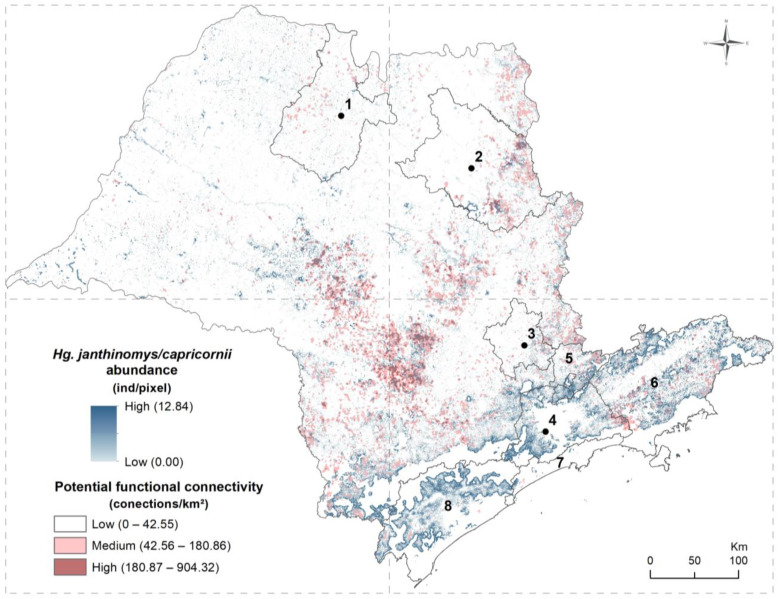
Estimated abundance and potential functional connectivity for *Haemagogus janthinomys*/*capricornii* in the state of São Paulo, indicating risk areas for mosquito dispersion. Numbered regions (1–4) correspond to the metropolitan regions of (1) São José do Rio Preto, (2) Ribeirão Preto, (3) Campinas, and (4) São Paulo, with the point indicating the centroid of the city that gives its name to the region; (5) corresponds to Serra da Cantareira; and (6–8) correspond to administrative regions: (6) Vale do Paraíba and Litoral Norte, (7) Baixada Santista, and (8) Vale do Ribeira.

**Table 1 ijerph-21-00609-t001:** Selected models (ΔAICc ≤ 2) to explain *Haemagogus leucocelaenus* (HL) and *Haemagogus janthinomys*/*capricornii* (HJ) abundances, according to the landscape structure (%FC—percentage of forest cover, %AG—percentage of agricultural land, %NV—percentage of native vegetation, NFF—number of forest fragments per km^2^, NNF—number of native-vegetation fragments per km^2^, FED—forest edge density, and NED—native-vegetation edge density), at different scales (values in parentheses in front of landscape variables, in kilometers (km)). DF—degrees of freedom.

Model	Response Variable	Predictor Variables	AICc	ΔAICc	DF	Weight of Evidence	Evidence Ratio
M1	HL	%FC (2 km)	75.5	0.0	3	0.164	1.0
M2	HL	%FC (1 km)	76.6	1.1	3	0.094	1.7
M3	HL	%FC (2.5 km)	76.7	1.2	3	0.091	1.8
M4	HL	%FC (1.5 km)	76.9	1.4	3	0.080	2.0
M5	HL	%FC (1 km) + NFF (1 km)	77.1	1.6	4	0.074	2.2
M6	HL	%VN (1 km) + NNF (1 km)	77.4	1.9	4	0.063	2.6
M1	HJ	%AG (2.5 km)	73.4	0.0	3	0.129	1.0
M2	HJ	%FC (2.5 km) + FED (2.5 km)	74.0	0.6	4	0.094	1.3
M3	HJ	%AG (2 km)	74.8	1.4	3	0.064	2.0
M4	HJ	%NV (2.5 km) + NED (2.5 km)	74.8	1.4	4	0.063	2.0

**Table 2 ijerph-21-00609-t002:** Sum of evidence weights of variables weighted by the number of models in which the variable appears (∑Wi/N), slope and standard error (±SD) for each variable present in selected models (ΔAICc ≤ 2) to explain *Haemagogus leucocelaenus* and *Haemagogus janthinomys*/*capricornii* abundances *(*%FC—percentage of forest cover, %AG—percentage of agricultural land, %NV—percentage of native vegetation, NFF—number of forest fragments per km^2^, NNF—number of native-vegetation fragments per km^2^, FED—forest edge density, NED—native-vegetation edge density, at different scales (values in parentheses in front of landscape variables, in kilometers (km)).

	*Hg. leucocelaenus*			*Hg. janthinomys*/*capricornii*		
Variables	N	∑Wi/N	Slope (±SD)	N	∑Wi/N	Slope (±SD)
%AG (2 km)				7	0.019	−0.032 (±0.010) *
%AG (2.5 km)				7	0.038	−0.034 (±0.010) *
%FC (1 km)	3	0.064	0.074 (±0.014) *			
%FC (1.5 km)	3	0.042	0.076 (±0.015) *			
%FC (2 km)	3	0.086	0.086 (±0.016) *			
%FC (2.5 km)	3	0.054	0.097 (±0.019) *	3	0.048	0.053 (±0.018) *
%NV (1 km)	3	0.044	0.077 (±0.016) *			
%NV (2.5 km)				3	0.026	0.043 (±0.019) *
FED (2.5 km)				3	0.042	−0.021 (±0.024)
NED (2.5 km)				3	0.032	−0.025 (±0.024)
NFF (1 km)	5	0.014	−0.032 (±0.251)			
NNF (1 km)	5	0.012	−0.347 (±0.266)			

* significant slope.

**Table 3 ijerph-21-00609-t003:** Sum of evidence weights for different scales for *Haemagogus leucocelaenus* and *Haemagogus janthinomys*/*capricornii*. The evidence ratio refers to the ratio between the weight of evidence of the best scale compared to that of the second best.

Species	0.5 km	1.0 km	1.5 km	2.0 km	2.5 km	Evidence Ratio
*Hg. Leucocelaenus*	<0.001	0.325	0.173	0.311	0.190	1.04
*Hg. janthinomys*/*capricornii*	0.031	0.092	0.100	0.251	0.504	2.00

## Data Availability

The vector abundance data is published and available at Mucci et al. [39,40].

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
