# Peer review of "Ecological Requirements for Abundance and Dispersion of Brazilian Yellow Fever Vectors in Tropical Areas"

_ijerph, 2024, doi:10.3390/ijerph21050609_

Round 1
Reviewer 1 Report
Comments and Suggestions for Authors
1)_ I think the authors should improve the introduction.
2) Statistical methods are poorly described. It is better to make the description of these methods a separate subsection.
3) When discussing the results, it is important to indicate the relationship between the spread of mosquitoes with different reservoirs or other water sources that are important for the development of larvae.
4) It is also important to understand the prospects for the further spread of mosquitoes in the landscape. Describe what practical results your research will yield in the future.
Author Response
Reviewer #1 (Remarks to the Author):
1. I think the authors should improve the introduction.
Authors: Thank you for this recommendation. We included some new information about the ecology of the vectors to better elucidate the justification for our hypothesis.
2. Statistical methods are poorly described. It is better to make the description of these methods a separate subsection.
Authors: Thank you for the suggestion. We provided additional information in the new version of the manuscript to better describe the methodology, as requested.
3. When discussing the results, it is important to indicate the relationship between the spread of mosquitoes with different reservoirs or other water sources that are important for the development of larvae.
Authors: Thank you very much for your comment. We included details in the Introduction section (lines 46 - 47) about the way the vectors reproduce, in order to clarify the raised issue. As the main oviposition and larvae development locations for these vectors are accumulated water in tree and bamboo holes, we considered we have indicated the relationship with the main water source for them, i.e, the forest.
4. It is also important to understand the prospects for the further spread of mosquitoes in the landscape. Describe what practical results your research will yield in the future.
Authors: Thank you for your suggestion. We included in the Conclusion section some information about how our work may help surveillance and epidemiological routines and analysis (lines 514 - 517).

Reviewer 2 Report
Comments and Suggestions for Authors
I have limited knowledge of the WYF disease system and the study area but my expertise is in landscape ecology and epidemiology. As far as I can judge, this study/manuscript is very well conceived and written. The goals and hypotheses are well defined, the methods are adequately chosen and applied, the results are clearly presented and interpreted in the context of existing literature. I have only minor suggestions:
The hypotheses 1 and 2 are clearly defined but could be better justified already in the Introduction. Only later in the manuscript the reader learns what the rationale is behind these hypotheses. I suggest to add a few more details in the Introduction that explain the reasons for their definition.
Lines 100-103: The authors state that they used the maximum radius of 2.5 km to avoid overlap among the buffers because the minimum distance between two collection points was 2.35 km. However, this would mean that there is quite a large overlap between the 2.5 km buffers of each of such pairs of nearby points. This would lead to high correlations in landscape metrics and potential issues with spatial autocorrelation (SAC) in the residuals of the statsitical models (GLMs). Did the authors test for SAC in their models? This should be reported in the paper.
One type of landscape metrics used in the study is the number of forest fragments and the number of native vegetation fragments. However, this is highly dependent on the study area (size of the radius). Was this metric really calculated as a "number" or a "number per area unit"? The earlier option would not allow meaningful comparison of the effect of this variable between different scales.
Author Response
Reviewer #2 (Remarks to the Author):
I have limited knowledge of the WYF disease system and the study area but my expertise is in landscape ecology and epidemiology. As far as I can judge, this study/manuscript is very well conceived and written. The goals and hypotheses are well defined, the methods are adequately chosen and applied, the results are clearly presented and interpreted in the context of existing literature.
Authors: Thank you very much for the positive comments and the following suggestions to improve our manuscript.
I have only minor suggestions:
1. The hypotheses 1 and 2 are clearly defined but could be better justified already in the Introduction. Only later in the manuscript the reader learns what the rationale is behind these hypotheses. I suggest to add a few more details in the Introduction that explain the reasons for their definition.
Authors: Thank you for your recommendation. We have provided additional information about the ecology of the vectors in the Introduction section, to better clarify the premises for our hypothesis.
2. Lines 100-103: The authors state that they used the maximum radius of 2.5 km to avoid overlap among the buffers because the minimum distance between two collection points was 2.35 km. However, this would mean that there is quite a large overlap between the 2.5 km buffers of each of such pairs of nearby points. This would lead to high correlations in landscape metrics and potential issues with spatial autocorrelation (SAC) in the residuals of the statistical models (GLMs). Did the authors test for SAC in their models? This should be reported in the paper.
Authors: Thank you very much for this important suggestion. We performed a Mantel test in this new version, to analyze the spatial autocorrelation, and for both species we detected that there is no spatial autocorrelation in our data. We described the analysis and the results in the Material and Methods section (lines 154 - 159).
3. One type of landscape metrics used in the study is the number of forest fragments and the number of native vegetation fragments. However, this is highly dependent on the study area (size of the radius). Was this metric really calculated as a "number" or a "number per area unit"? The earlier option would not allow meaningful comparison of the effect of this variable between different scales.
Authors: Thank you for your comments. Indeed, we were aware of this issue, however, as our goal was to compare landscapes among sample points, not among different scales, we have chosen to use the mentioned variable without pondering by area. Nonetheless, after your observations, we reanalyzed the data aiming to address the raised issue. Although most of the results have remained the same or very similar to the previous ones, one more model was selected for Haemagogus leucocelaenus. Then, we presented the results of this new analysis into the revised manuscript.
